# Magneto-Fluorescent Hybrid Sensor CaCO_3_-Fe_3_O_4_-AgInS_2_/ZnS for the Detection of Heavy Metal Ions in Aqueous Media

**DOI:** 10.3390/ma13194373

**Published:** 2020-09-30

**Authors:** Danil A. Kurshanov, Pavel D. Khavlyuk, Mihail A. Baranov, Aliaksei Dubavik, Andrei V. Rybin, Anatoly V. Fedorov, Alexander V. Baranov

**Affiliations:** Center of Information Optical Technology, ITMO University, 49 Kronverksky Prospekt, 197101 St. Petersburg, Russia; kurshanov.danil@gmail.com (D.A.K.); khavlyuk.stepnogorsk@gmail.com (P.D.K.); mbaranov@mail.ru (M.A.B.); adubavik@itmo.ru (A.D.); rybin@mail.ifmo.ru (A.V.R.); a_v_fedorov@inbox.ru (A.V.F.)

**Keywords:** magnetic–luminescent structure, hybrid system, ternary quantum dots, magnetic nanoparticles, iron oxide, calcium carbonate microspheres, sensor

## Abstract

Heavy metal ions are not subject to biodegradation and could cause the environmental pollution of natural resources and water. Many of the heavy metals are highly toxic and dangerous to human health, even at a minimum amount. This work considered an optical method for detecting heavy metal ions using colloidal luminescent semiconductor quantum dots (QDs). Over the past decade, QDs have been used in the development of sensitive fluorescence sensors for ions of heavy metal. In this work, we combined the fluorescent properties of AgInS_2_/ZnS ternary QDs and the magnetism of superparamagnetic Fe_3_O_4_ nanoparticles embedded in a matrix of porous calcium carbonate microspheres for the detection of toxic ions of heavy metal: Co^2+^, Ni^2+^, and Pb^2+^. We demonstrate a relationship between the level of quenching of the photoluminescence of sensors under exposure to the heavy metal ions and the concentration of these ions, allowing their detection in aqueous solutions at concentrations of Co^2+^, Ni^2+^, and Pb^2+^ as low as ≈0.01 ppm, ≈0.1 ppm, and ≈0.01 ppm, respectively. It also has importance for application of the ability to concentrate and extract the sensor with analytes from the solution using a magnetic field.

## 1. Introduction

The industrial development has resulted in constantly increasing levels of heavy metal contamination in the environment [1,2]. To reduce environmental pollution, it is necessary to determine heavy metal ions in soil [3,4] and water resources [5,6]. The detection and selective quantitative definition of heavy metal ions in nature-conservation resources or biological samples have been an important research area for a long time. The development and creation of sensors based on nanoparticles (NPs) have experienced significant growth over the past decades [7,8,9].

Traditional methods for heavy metal analysis include atomic absorption spectrometry [10,11], inductively coupled plasma mass spectrometry [12,13], and different electrochemical analyses, e.g., potentiometric techniques [14]. However, these methods have complex sample preparation processes, low stability, and are not compatible with different environments, which significantly limits their applicability.

An alternative method is optical detection. The optical research via fluorescence or colorimetric is the most convenient and hopeful method due to its simplicity and high sensitivity of detection [15,16]. In this area, the application of ion-sensitive luminescent quantum dots (QDs) has been of special interest because of different photophysical properties, customizable structures, and the ability to bind to various ligands, and the possibility integrates into the various hybrid systems with different functional properties while conserving their luminescence properties [17,18,19].

Many QDs sensor systems are based on binary cadmium compounds, which are hydrophobic toxic compounds [20,21,22]. Alternative Сd-free nanomaterials are QDs with ternary compositions based on silver or copper, AgInS_2_, and CuInS_2_ QDs, which also possess a set of physicochemical properties [23,24,25,26] necessary for creating luminescent sensors of toxic compounds. Their photoluminescent (PL) properties are characterized by high quantum PL yield (QYs) ≥ 50% in the visible region from 400 to 1000 nm in the near-infrared (NIR) region after shell growth with, for example, a ZnS semiconductor [24]. Another distinctive characteristic of ternary QDs is broad PL bands with a full width at half-maximum (FWHM) from 100 nm, which slightly overlaps the QD absorption spectrum (Stokes shifts ≈1 eV [27]), in which there is no sharp first exciton peak [28]. Moreover, ternary QDs show long PL lifetimes of some hundred nanoseconds [27,28]. Previous studies demonstrate the possibility of using AgInS_2_ and CuInS_2_ QDs in methods for detecting heavy metal ions [29,30,31].

In addition to detection, it is also important to remove heavy metal ions from the contaminated environment. One of the cleaning methods is sorption using various mesoporous materials, for example, Ca-based materials [32]. Calcium carbonate (CaCO_3_) crystals are widely used for the manufacturing of carriers containing various embedded nanoparticles or biologically active compounds [33,34,35]. Nowadays, these CaCO_3_ crystals (vaterite form) are one of the most popular approaches for matrix formation due to their easy synthesis [36,37], low dispersion, control over crystal size (in the micrometer and submicrometer range [38]), and spherical shape [39], which makes these crystals an attractive candidate for many applications [38,39].

In this work, we focus on the design of AgInS_2_-based sensors for the most common toxic heavy metal ions, Co^2+^, Ni^2+^, and Pb^2+^, which can accumulate in the human body and cause acute or chronic diseases [40,41]. The sensor is a hybrid complex based on a matrix of porous CaCO_3_ microspheres doped by Fe_3_O_4_ magnetic nanoparticles. The surface of the spheres is covered by a shell of several layers of polyelectrolytes. This system combines the properties of several different components, such as the absorption properties of CaCO_3_ microspheres [42], the photoluminescent properties of AgInS_2_/ZnS QDs, and the magnetic properties of Fe_3_O_4_ nanoparticles.

## 2. Materials and Methods

### 2.1. Materials

All reagents purchased from Sigma-Aldrich, Steinheim, Germany, were used without further purification. In all procedures, deionized Hydrolab water was used.

To synthesize AgInS_2_/ZnS QDs, we used indium(III) chloride (InCl_3_), silver nitrate (AgNO_3_), zinc(II) acetate dihydrate (Zn(Ac)_2_), sodium sulfide (Na_2_S·9H_2_O), an aqueous solution of ammonia hydrate (NH_4_OH), thioglycolic acid (TGA), and isopropyl alcohol.

To synthesize Fe_3_O_4_ magnetic nanoparticles, we used (tris(acetylacetonato)iron(III)), triethylene glycol (TEG), tetrahydrofuran (THF).

To synthesize CaCO_3_ microspheres, we used sodium carbonate (Na_2_CO_3_), calcium chloride (CaCl_2_), poly(sodium 4-styrenesulfonate) sodium salt (PSS, Mw = 70 kDa) and poly(allylamine hydrochloride) (PAH, Mw = 70 kDa).

The aqueous solutions of heavy metals were prepared by dissolving the salts of metals (cobalt(II) nitrate, nickel(II) sulfate, lead(II) chloride) in water.

### 2.2. Methods

#### 2.2.1. Synthesis of Fe_3_O_4_ Nanoparticles

The magnetic Fe_3_O_4_ nanoparticles were prepared by mixing iron (III) triacetylacetonate and TEG. In the synthesis, 1 mmol of iron precursor and 24 mL of triethylene glycol were added to a three-necked flask under magnetic stirring. The mixture was degassed at up to 90 °C and kept for 60 min under vacuum. After degassing and flushing with argon, the solution was heated to 275 °C and kept for 2 h under a constant Ar flow. After the synthesis, the NCs were washed with THF by centrifugation and then dissolved in water for storage.

#### 2.2.2. Synthesis of AgInS_2_/ZnS QDs

For the synthesis of AgInS_2_ quantum dot cores, 1 mL of AgNO_3_ water solution (0.1 М), 2 mL of TGA water solution (1.0 М), and 0.2 mL of NH_4_OH (5.0 М) water solution were added to 92 mL of water under magnetic stirring and ambient conditions. Then, 0.45 mL of NH_4_OH solution (5.0 М) and 0.9 mL of InCl_3_ water solution (1.0 М) containing 0.2 М HNO_3_ were added. After that, the solution changed its color from yellow to colorless. After adding 1 mL of 1.0 М Na_2_S water solution (1.0 М), the resultant solution was heated at 95 °С for 30 min by a water bath. For ZnS shell growth on the surface of AgInS_2_, 1 mL of TGA solution (1.0 М) and 1 mL of Zn(Ac)_2_ solution (1.0 M) containing 0.01 М HNO_3_ was added.

After the synthesis, the AgInS_2_/ZnS quantum dot solution was cooled and concentrated using rotary evaporation. For the size-selection procedure, the aggregation of quantum dots was initiated by adding 0.5 mL of isopropyl alcohol and subsequent centrifugation at 10,000 rpm for 5 min. The precipitate was separated and marked further as fraction #1. This procedure was repeated until the solution was fully discolored. Here, we used quantum dots with a luminescence peak at ≈600 nm.

#### 2.2.3. Preparation of CaCO_3_-Fe_3_O_4_-AgInS_2_/ZnS (CFA) Fluorescent Sensor

Na_2_CO_3_ (0.33 M; 700 μL), CaCl_2_·2H_2_O (0.33 M; 700 μL), and Fe_3_O_4_ nanoparticles (50 μL concentrated aqueous solution) were added to a round-bottom flask. The resulting solution was stirred for 30 s, after which the resulting spheres were centrifuged for 40 s at 3000 rpm. Then, precipitated CaCO_3_-Fe_3_O_4_ microspheres were washed twice with H_2_O.

The next step was to extend the shell using the Layer-by-Layer (LbL) method. First, 1 mL of PAH solution (6 mg/mL, 0.5 M NaCl, and pH 6.5) was added to the precipitated CaCO_3_-Fe_3_O_4_ spheres. The resulting dispersion was shaken for 10 min. Excess polyelectrolyte was removed by washing with water and centrifuging (30 s at 4000 rpm). The procedure was repeated twice. Then, the procedure was repeated using PSS solution (6 mg/mL, 0.5 M NaCl, and pH 6.5). After triple coating with polymer layers (PAH/PSS/PAH), 100 μL of a QDs stock solution was added to the spheres. The dispersion was shaken for 10 min. The excess QDs, by analogy with polyelectrolytes, were centrifuged and washed with water. After the QDs layer, a double layer of PSS and PAH was applied to the spheres.

The resulting fluorescent sensors based on CaCO_3_ microspheres were dispersed in water.

### 2.3. Equipments

A spectrophotometer UV-3600 (Shimadzu, Kyoto, Japan) and spectrofluorometer FP-8200 (Jasco, Tokyo, Japan) were used for recording the absorption and PL spectra of the samples, respectively. The SEM and STEM images of the studied samples were taken with a Merlin (Zeiss, Oberkochen, Germany) scanning electron microscope while a FEI Titan electron microscope operating at a voltage of 300 kV was used for getting the TEM image of Fe_3_O_4_ nanoparticles. The transmitted light images were obtained with a LSM-710 (Zeiss, Oberkochen, Germany) laser scanning confocal microscope equipped with a microobjective of NA = 0.95. The photoluminescence (PL) decay curves of the sample were obtained with a laser scanning confocal microscope MicroTime 100 (PicoQuant, Berlin, Germany) equipped with a pulsed light source with the 80 ps pulses at a repetition rate of 0.2 MHz and a Time Correlated Single Photon Counter (TCSPC) detector. The size of magnetic nanoparticles was determined by dynamic light scattering (DLS) using a Malvern Zetasizer Nano (Malvern, Worcestershire, UK).

## 3. Results

The formation of the CFA fluorescent sensor was produced (made) in two steps (scheme Figure 1a). In the first step, the CaCO_3_-Fe_3_O_4_ microspheres have been formed by using concentrated aqueous solutions of Na_2_CO_3_ and СaCl_2_ and magnetic Fe_3_O_4_ nanoparticles of 5–6 nm mean size. Magnetic Fe_3_O_4_ nanoparticles are located inside the pores of CaCO_3_ spheres providing a brownish color of CaCO_3_-Fe_3_O_4_. The SEM (STEM) images of CaCO_3_ microspheres, doped by Fe_3_O_4_ nanoparticles, Fe_3_O_4_ magnetic nanoparticles, and AgInS_2_/ZnS QDs stabilized with TGA are shown in Figure 1b–d, respectively.

The resulting CFA microspheres with polyelectrolytic shell PAH/PSS were characterized by scanning electron microscopy, DLS, UV-Vis, and photoluminescent steady-state and transient microscopy. Their SEM images are presented in Figure 1e and show spherical microparticles with a mean size of ≈4 μm. DLS measurements showed that an average hydrodynamic diameter of the CaCO_3_-Fe_3_O_4_ microsphere is about 3.5–4 μm as determined from the analysis of the size-distribution profile (Figure 1e, inset).

The next step was the formation of polyelectrolyte multilayers and the deposition of the QDs layer on a surface of the microspheres by LbL method. Oppositely charged electrostatically interacting polymers and QDs have deposited alternately on the surface of microspheres [43,44,45]. In this work, we used PAH and PSS polymers and QDs AgInS_2_/ZnS core/shell capped with the hydrophilic ligand TGA. To heighten the PL properties and stability of the AgInS_2_ cores, the cores were passivated by a protective ZnS shell. The resulting stable solution QDs AgInS_2_/ZnS core/shell had a mean size of ≈5 nm and with a PL QY 30%. At first, two double layers of PAH/PSS polyelectrolytes were deposited on the surface of CaCO_3_-Fe_3_O_4_; then, AgInS_2_/ZnS QDs were applied over the first layers of the polyelectrolytes shell by electrostatic adsorption using the LbL method. Finally, an additional protective PAH/PSS double layer was placed on the microsphere surface. The resulting CFA fluorescent sensor is illustrated in Figure 1e.

The optical and magnetic properties of the CFA microspheres are illustrated in Figure 2.

The magneto-fluorescent properties of CFA allow easily concentrating sensors with a magnet or detect their PL under ultraviolet light, as shown in Figure 2a.

The absorption spectra of the AgInS_2_/ZnS QDs, the CaCO_3_-Fe_3_O_4_ microspheres, and fluorescent CFA sensors are shown in Figure 2b. It is seen that the absorption spectrum of CaCO_3_-Fe_3_O_4_ microspheres contains a noticeable contribution from elastic scattering, which is caused by the rather large size of the microspheres. The absorption of AgInS_2_/ZnS QDs is characterized by a broad and peak-less spectrum with the absorbance increasing gradually in the shortwave region. The absorption spectra of CaCO_3_-Fe_3_O_4_ microspheres and AgInS_2_/ZnS QDs additively contribute to the absorption spectrum of the CFA microspheres.

The PL spectra of the CFA microspheres and the AgInS_2_/ZnS QDs in aqueous solution are shown in Figure 2b. This PL spectrum demonstrates a broad and symmetric luminescence band in the visible and NIR region, similar to that of the AgInS_2_/ZnS QDs in water. Inclusion of the AgInS_2_/ZnS QDs into polymer layers on the surface of CaCO_3_-Fe_3_O_4_ microspheres leads to a small redshift of the PL band as compared to that in solution, which is most likely due to the interaction between close QDs. Confocal transmitted light and FLIM images of the CFA microspheres presented in Figure 2c,d, respectively, show that luminescent QDs are embedded in the surface layer of the microspheres of about 3–4 μm in diameter, demonstrating bright PL response.

As in the case of the AgInS_2_/ZnS QDs in the solution, the PL decay of the QDs in the CFA microspheres is characterized by multiexponential decay kinetics, which is a characteristic property of the PL of ternary QDs that originates from the complicated structure of the low-energy states of the AgInS_2_ QDs [46]. At the same, an average PL lifetime, calculated by the formula:<τ> = ΣA_i_τ_i_^2^/ΣA_i_τ_i_,
where A_i_ is the amplitude and τ_i_ is the decay time of the i-th exponent, which for the embedded QDs of about 200 ns is remarkably smaller than that of 350 ns for the QDs in aqueous solution. This difference that is seen in Figure 2e shows that embedding QDs in the surface layer of microspheres results in the appearance of the nonradiative channel of the PL decay.

To demonstrate the ability of our fluorescent sensor for the optical detection of heavy metal ions Co^2+^, Ni^2+^, and Pb^2+^, aqueous metal solutions with a concentration of 0.001 M were prepared. To the CFA solution with a volume of 100 μL per 3 mL of water, metal salt solutions were added with a volume of 3 to 300 μL. Then, the optical properties of the CFA in the presence of heavy metal ions were studied.

The results presented in Figure 3 show that the photoluminescence intensity decreases with an increase in the concentration of heavy metals ions. It is not surprising, since the core–shell AgInS_2_/ZnS QDs has the stabilizer of thioglycolic acid, which forms the negatively charged layer on the surface of the nanocrystals. The quenching of the PL is observed due to the Coulombic interaction of positively charged metal ions from the analyte with the negative organic shell [47]. Data on the ion concentration dependence of the QD PL are reproduced in three independent experiments, which are reflected in values of corresponding PL intensity measurement errors in the right panels of Figure 3. With low concentrations of Co^2+^ ions, a significant decrease in the luminescence intensity of the sensor is observed (Figure 3a). These suggest a high sensitivity of the fluorescent sensor for Co^2+^ ions. The sensitivity of the sensor for Ni^2+^ and Pb^2+^ was lower by an order of magnitude (Figure 3b,c, respectively). Inserts in Figure 3 (right panel) show in detail the measured PL reduction with the addition of the smallest amount of the Co^2+^, Ni^2+^, and Pb^2+^ ions. A simple estimation of low limits of ion concentrations in solution by measuring the reduction of the sensor PL intensity as compared with the ion-free solution within experimental errors of ≈1–2%, gives concentrations of 10^−8^ M Co^2+^ and 10^−7^ M Ni^2+^ and Pb^2+^ even without optimization of the sensor parameters. These values are close to the detection limit reported in [48] for a dissociative CdSe/ZnS QD/PAN complex for luminescent sensing of metal ions in aqueous solutions and orders of magnitude higher for Co(II) and for Ni(II) obtained by colorimetric (absorption) measurements with 1-(2-pyridilazo)-2-naphtol (PAN) as a complexing reagent in the aqueous phase using the non-ionic surfactant Tween 80 [49]. The difference in the luminescence extinguishing of ternary quantum dots by Co^2+^, Ni^2+^, and Pb^2+^ is not quite understood at the moment and is the subject of consideration.

At the same time, it should be noted that despite the good sensitivity, the proposed fluorescent sensor does not exhibit selectivity for ions recognition. Therefore, its practical use is limited by a quick on-line preliminary analysis of the presence of heavy metal ions in water samples and a recommendation for further detailed analysis of the elemental composition of ions.

Due to the presence of magnetic Fe_3_O_4_ nanoparticles in the microspheres, the sensor with absorbed metal ions can be removed from the solution using a magnetic field, for example, for further analysis of the chemical and/or elemental composition of toxic impurities. To demonstrate this phenomenon, the 20 μL of a 0.001 M aqueous solution of Co^2+^ ions was added to the sensor based on CFA microspheres. In this case, a decrease up to 35% of the initial value was observed in the intensity of the photoluminescence of the sensor (Figure 4b). After the sensor was removed from the test solution using a magnet, the purified solution was again analyzed using the CFA sensor at the same amount (Figure 4c–d). The intensity of luminescence was practically unchanged compared to the solution without metal ions and it decreased by 15%, which indicates a significant decrease in the content of metal ions. This result suggests that the microspheres were successfully absorbed by the heavy metal ions Co^2+^ in solution.

This result suggests that the microspheres successfully absorb the heavy metal ions in solution, which might be concentrated by the magnetic field e.g., further analytical and (micro)biological applications in flow cytometry.

## 4. Conclusions

In this work, we obtained a sensor based on a porous matrix, which is microspheres of calcium carbonate doped by magnetic Fe_3_О_4_ nanoparticles and luminescent AgInS_2_/ZnS QDs. The magnetic properties of Fe_3_О_4_ and photoluminescent properties of QDs AgInS_2_/ZnS were combined, yielding highly magnetic and luminescent calcium carbonate microspheres as magneto-fluorescent sensors of heavy metal ions in aqueous solutions. To demonstrate the potential of sensors for sensitivity to in heavy metal ions, the microspheres CaCO_3_-Fe_3_O_4_-AgInS_2_/ZnS were placed in aqueous solutions containing Co^2+^, Ni^2+^, and Pb^2+^ ions with various concentrations. By measuring the quenching the PL of the sensor in the presence of heavy metal ions, we demonstrated the possibility of metal ions detection down to a concentration of 10^−8^ M Co^2+^ and 10^−7^ M Ni^2+^, and Pb^2+^ (≈0.01 ppm, ≈0.1 ppm, and ≈0.1 ppm, respectively). 

The main distinguishing feature of the proposed sensor from the luminescent sensors previously reported, e.g., [50,51] is the presence of magnetic properties; therefore, in this work, we also demonstrate the ability to concentrate and remove the sensor from the analyzed system together with ions of the analyzed metals for their precise identification, for example, by the inductively coupled plasma mass spectrometry. These concentrations of ions, which could cause a statistically significant quenching of sensors PL, are much lower than their maximum allowable concentration in natural water. It opens prospects of CaCO_3_-Fe_3_O_4_-AgInS_2_/ZnS sensors for developing methods for the environmental monitoring of heavy metal ions.

## Figures and Tables

**Figure 1 materials-13-04373-f001:**
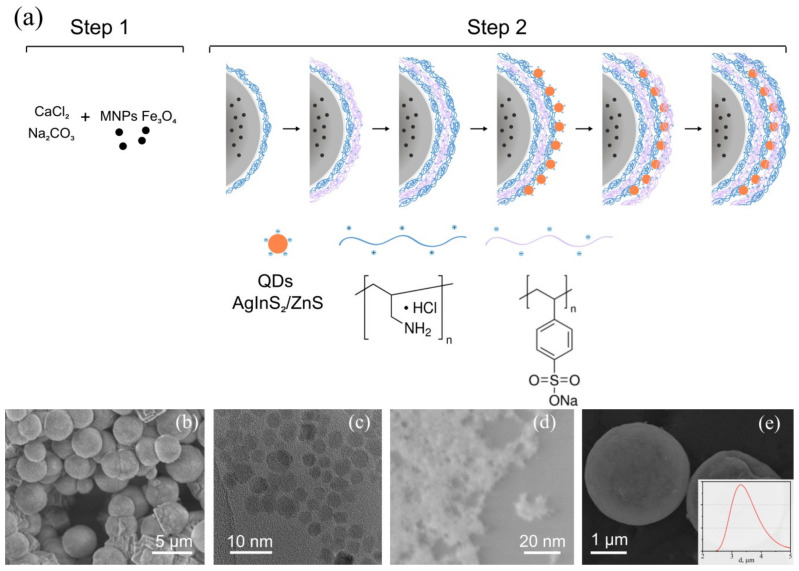
(**a**) Scheme of a two-stage synthesis of a magnetic luminescent sensor based on CaCO_3_ microspheres, (**b**−**e**) SEM and STEM images of the sensor components: (**b**) CaCO_3_ microspheres, doped by Fe_3_O_4_ nanoparticles, (**c**) Fe_3_O_4_ magnetic nanoparticles, (**d**) AgInS_2_/ZnS quantum dots (QDs) stabilized with thioglycolic acid (TGA), (**e**) CFA (CaCO_3_-Fe_3_O_4_-AgInS_2_/ZnS) microspheres with polyelectrolyte shell poly(allylamine hydrochloride)/poly(sodium 4-styrenesulfonate) (PAH/PSS). Inset in (**e**) shows a typical dynamic light scattering (DLS) size-distribution plot.

**Figure 2 materials-13-04373-f002:**
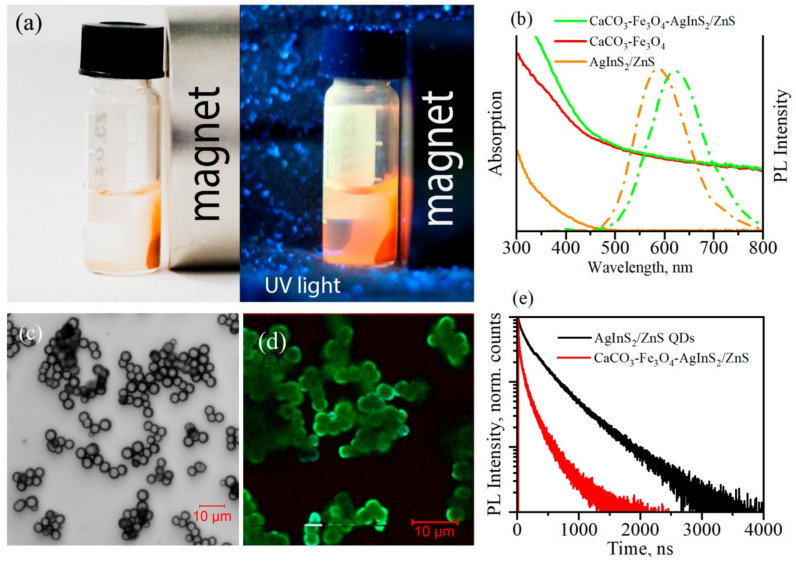
Optical and magnetic properties of the CFA fluorescent sensor. (**a**) Photos of the accumulation of the microspheres to magnetic bar under UV light, (**b**) Absorption (solid line) and photoluminescent (PL) (dash/dotted line) spectra of the CFA fluorescent sensor and AgInS_2_/ZnS QDs in water (orange AgInS_2_/ZnS QDs, red CaCO_3_-Fe_3_O_4_ microspheres, and green CFA fluorescent sensor), (**c**) Confocal transmitted light images of the CFA microspheres, (**d**) Fluorescence-Lifetime Imaging Microscopy (FLIM) images of the CFA microspheres, (**e**) PL decay curves of the CFA microspheres and the aqueous dispersion of the initial AgInS_2_/ZnS QDs. The PL measurements shown in panels (**b**−**e**) were done at 350 nm excitation.

**Figure 3 materials-13-04373-f003:**
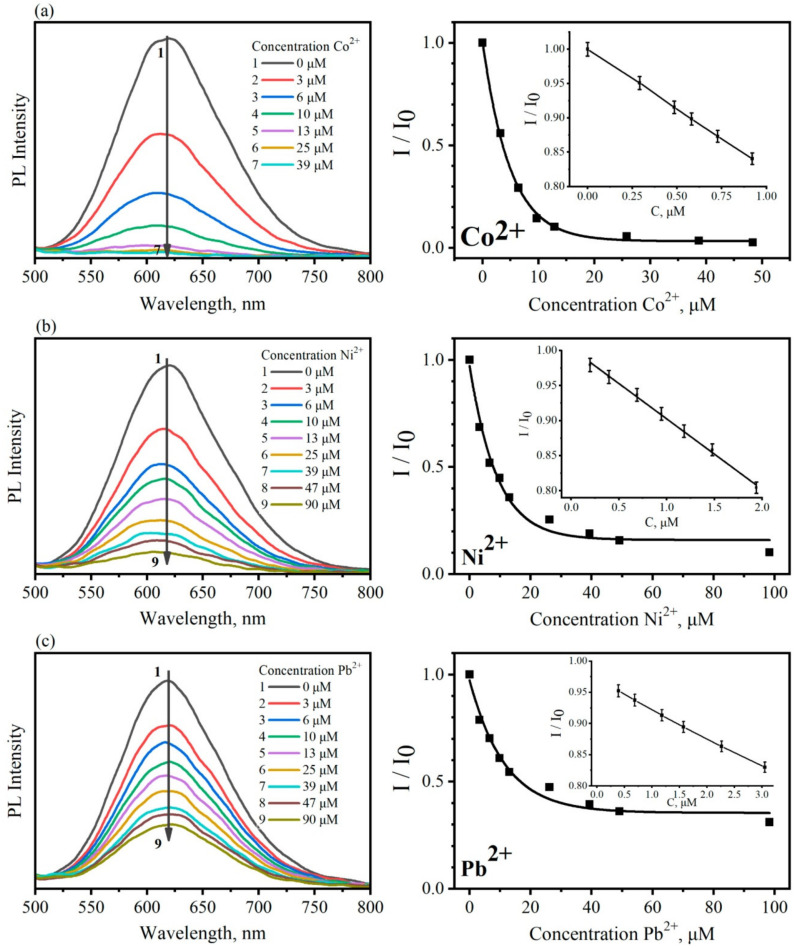
Quenching the CFA sensor luminescence upon increasing concentration heavy metal ions in aqueous solution. Left panel—PL band of CFA sensors as a function of the concentration of the Co^2+^, Ni^2+^, and Pb^2+^ ions. Right panel—integrated luminescence intensity ratio (I_0_/I) dependence on the concentration of heavy metal ions for (**a**) Co^2+^, (**b**) Ni^2+^, (**c**) Pb^2+^; insets show the regions of low ions concentration.

**Figure 4 materials-13-04373-f004:**
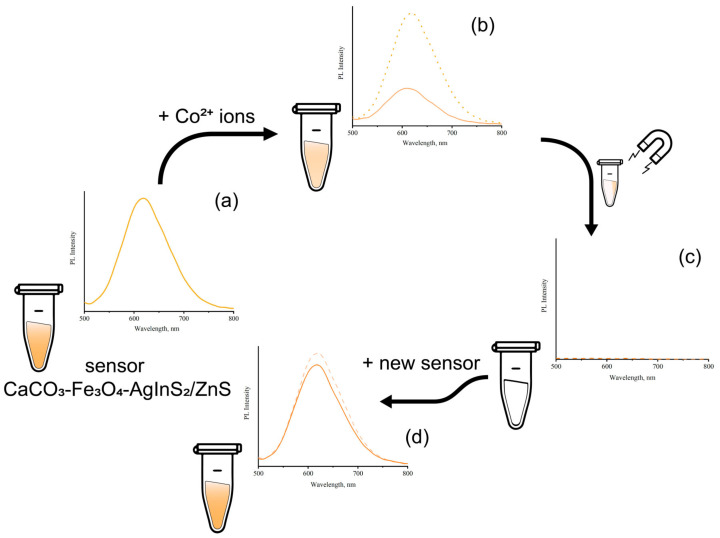
(**a**) PL spectra of the CFA fluorescent sensor without heavy metal ions, (**b**) PL spectra of the CFA fluorescent sensor with Co2+ ions in comparison with a solution without the presence of heavy metal ions (dashed line), (**c**) PL spectra of the solution after removing the CFA fluorescent sensor, (**d**) PL spectra of the solution after adding a new portion of CFA sensor in comparison with a solution without the presence of heavy metal ions (dashed line).

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
