# Peer review of "Magneto-Fluorescent Hybrid Sensor CaCO3-Fe3O4-AgInS2/ZnS for the Detection of Heavy Metal Ions in Aqueous Media"

_materials, 2020, doi:10.3390/ma13194373_

Round 1

Reviewer 1 Report

The manuscript by Kurshanov et al. deals with a heavy metal ion sensing system in aqueous medium based on CaCO3-Fe3O4-AgInS2/ZnS (CFA) fluorescent sensor. The sensor is capable of detecting Co2+, Ni2+ and Pb2+. Finally, the authors demonstrated removal of the sensor with the analyte from the solution by magnetic field.

Although the topic is of current importance, the system design sensing results or detection limits claimed by the authors are not very convincing. Therefore, I would not recommend the publication of the manuscript in materials.

Specific comments:

Introduction lacks in sufficient discussion of the existing similar sensing systems.

In which respect the present system is novel compared to the other available  systems?

In the introduction, the authors described Co2+, Ni2+ and Pb2+ as the 'most relevant toxic heavy metal ion'. What do they mean by most relevant, please clarify.

Line 139: The authors referred to the DLS result but I couldn't see relevant data included in the manuscript.

Fig.2b: The labeling is incomplete and confusing. Please address.

Line 186: Referring to Fig.3a authors stated that the decrease in luminescence intensity in presence of Co+ is significant. Without a quantitative data such observation doesn't proof any high detection/detection limit.

Line 189~191: Detection limit of the various heavy metal ions is very important in the present study. Therefore, it is very important to know exactly how did authors actually measured the detection limit need to be added. Authors need to clarify their claim.

Please discuss why the sensitivities of the sensor system towards Co2+, Ni2+, Pb2+ are different.

Fig.3: Legend/label are incomplete.

Line 206: Authors indicated only 'slight change' in luminescence intensity. However, at the same time they interpreted that the the associated change was indicative of significant decrease in metal content.  They sound very contradictory, and therefore need to be clarified.

In the conclusion the authors claimed that they obtained an 'universal sensor'. In what respect their system can be called universal?

Author Response

Dear Reviewer,
Thank you very much for the valuable comments, remarks, and suggestions. We
have modified the manuscript according to the comments below. We have also
corrected typos and language style and slightly restructured the text of the
manuscript for the ease of reading. The reviewer comments in this response are
marked by bold. The revisions have been marked in this response and in the
manuscript by color text. We hope this improved the manuscript and it will be
suitable for publication in Materials.

Reviewer 2 Report

The manuscript “Magneto-fluorescent hybrid sensor CaCO3-Fe3O4 AgInS2/ZnS for the detection of heavy metal ions in aqueous media” presents an experimental study of a sensor to detect heavy metals.

Overall, the manuscript is very clear and well written, and the results are supported by the data.

Nevertheless, I have few comments.

  • Some error bars are missing on figure 3.
  • The sensing mechanism is reproducible? Some discussion about this subject is missing.
  • I did not see any information about the specificity. It should be a general sensor for heavy metals, or it is possible to identify? I think that discussion is very important.

Author Response

(The authors gave the same response as above.)

Reviewer 3 Report

In the present manuscript, magnetofluorescent microspheres have been prepared and used as sensors for the detection of heavy metal ions. The idea and topic are interesting. However, the overall quality of the manuscript is low. Below are some of my comments.

1) PL decays are reported but no details are given. The authors are encouraged to provide specific lifetimes as well as attribution of these to physical mechanisms.

2) Figures 2c and d are not discussed in the manuscript.

3) The authors should provide at least some mechanistical model on how the quenching of the PL of the microsphers is realized because of the metal ions. This will provide the reader with more physical insight. 

4) The authors provide results of the detection of three different metal ions by their sensors via a PL quenching. For all three metal ions, a qualitative similar PL quenching is observed. So, how one can identify which one of the ions is present? Therefore, my main concern is that their sensors do not have selectivity. The sensors may experience similar PL quenching even with other ions. A sensor should have accuracy, repeatability and selectivity. That is, it should detect a specific metal ion and should not be sensitive to any other ion. 

5) Finally, there are several sentences which do not make sense due to low-quality of the writting. Here are some examples.

Alternative Сd-free nanomaterials are QDs having ternary compositions based on silver or copper, Ag–In–S (AIS) and Cu–In-S (CIS) QDs, that also show size confinement effects [23], are particularly interesting

First, the CaCO3 microspheres were synthesized using concentrated aqueous solutions of Na2CO3 and СaCl2 as well as magnetic Fe3O4 nanoparticles of 5-6 nm mean size for the formation of the porous CaCO3-Fe3O4 microspheres.

The next step was the formation of a and the deposition

The absorption spectra of the AgInS2/ZnS QDs and resulting fluorescent CFA sensors with the extinction spectrum are presented in Figure 2b.

The presence of magnetic Fe3O4 nanoparticles in the microspheres, the sensor with absorbed metal ions can be removed from the solution using a magnetic field

Author Response

(The authors gave the same response as above.)

Reviewer 4 Report

The manuscript has been prepared with good information.  Samples and data have been used in a properly way. Therefore, I would like to mention some points about the aforementioned paper in order to be taken into account by the authors:

  1. Page 6, line 195 the authors mention that “…for Co(II) and for Ni(II) obtained by colorimetric measurements with PAN…”.

Please give more details about the colorimetric method used for this determination of metals.

  1. Did the authors measured residual concentrations of heavy metals or they based only on the fact that the photoluminescence intensity decreases with an increase in concentration of heavy metals ions?

  1. The problem with the analysis of the results is that are not compared with the existing literature. Please provide relevant comparisons. The section 3. Results need to be further discussed, especially in the part concerning the removal of metals.

Author Response

(The authors gave the same response as above.)

Round 2

Reviewer 1 Report

The authors mostly addressed the comments in the revised version. I would recommend the acceptance once the folowing issues are addressed.

DLS: I still belive DLS size-distribution plot is necessary to include atleast in the supporting info if not in the main manuscript.

Line 65: It should be 'metal ions' not 'metal ion'. I would recommend the authors to check entire manuscript for correctness.

Author Response

Dear Reviewer,

Thank you very much for the valuable comments and suggestions. We have modified the manuscript according to the comments below. We have also corrected some typos. The reviewer comments in this response are marked by bold. The revisions have been marked in this response and in the manuscript by color text. We hope this improved the manuscript, and it will be suitable for publication in Materials.

Reviewer 3 Report

The authors have addressed most of the comments especially the one regarding the selectivity of the sensors. The manuscript can now be accepted.

Author Response

Dear Reviewer,

Thank you very much for the valuable comments, remarks, and suggestions. We are grateful for your help in improving the manuscript. We carefully checked the entire text of the article again and have corrected some types.
